# Selective role of the DNA helicase Mcm5 in BMP retrograde signaling during *Drosophila* neuronal differentiation

**Irene Rubio-Ferrera**[1], **Pablo Baladrón-de-Juan**[1], **Luis Clarembaux-Badell**[1], **Marta Truchado-Garcia**[2], **Sheila Jordán-Álvarez**[1], **Stefan Thor**[3], **Jonathan Benito-Sipos**[1]*, **Ignacio Monedero Cobeta**[1,4]*

**1** Departamento de Biología, Facultad de Ciencias, Universidad Autónoma de Madrid, Cantoblanco, Madrid, Spain, **2** University of California Berkeley, Berkeley, California, United States of America, **3** School of Biomedical Sciences, The University of Queensland, Brisbane, Australia, **4** Departamento de Fisiología, Facultad de Medicina, Universidad Autónoma de Madrid, Cantoblanco, Madrid, Spain

* jonathan.benito@uam.es (JB-S); ignacio.monedero@uam.es (IMC)

**Data Availability Statement:** All relevant data are within the manuscript and its Supporting Information files.

## Abstract

The MCM2-7 complex is a highly conserved hetero-hexameric protein complex, critical for DNA unwinding at the replicative fork during DNA replication. Overexpression or mutation in MCM2-7 genes is linked to and may drive several cancer types in humans. In mice, mutations in MCM2-7 genes result in growth retardation and mortality. All six MCM2-7 genes are also expressed in the developing mouse CNS, but their role in the CNS is not clear. Here, we use the central nervous system (CNS) of *Drosophila melanogaster* to begin addressing the role of the MCM complex during development, focusing on the specification of a well-studied neuropeptide expressing neuron: the Tv4/FMRFa neuron. In a search for genes involved in the specification of the Tv4/FMRFa neuron we identified *Mcm5* and find that it plays a highly specific role in the specification of the Tv4/FMRFa neuron. We find that other components of the MCM2-7 complex phenocopies *Mcm5*, indicating that the role of *Mcm5* in neuronal subtype specification involves the MCM2-7 complex. Surprisingly, we find no evidence of reduced progenitor proliferation, and instead find that *Mcm5* is required for the expression of the type I BMP receptor Tkv, which is critical for the FMRFa expression. These results suggest that the MCM2-7 complex may play roles during CNS development outside of its well-established role during DNA replication.

## Author summary

The MCM2-7 complex plays a critical role in the DNA replication allowing cells to progress throughout the cell cycle and divide. Overexpression or mutation in MCM2-7 genes is linked to and may drive several cancer types in humans. While MCM2-7 complex is widely expressed in the central nervous system (CNS) during development, its role is not yet clear. Here, we use the CNS of *Drosophila melanogaster* to address the role of the MCM complex, focusing on the specification of a well-studied neuropeptide expressing neuron: the Tv4/FMRFa neuron. We identified that *Mcm5* plays a highly specific role in

**Funding:** This work was supported by a grant from the Misterio de Ciencia y Educación (PID2019-110952GB-I00) to J.B-S, and The University of Queensland, Australia, to ST. The funders had no role in study design, data collection and analysis, decision to publish, or preparation of the manuscript.

**Competing interests:** The authors have declared that no competing interests exist.

the specification of this neuron, and it involves other components of the MCM2-7 complex. Despite the described importance of this complex on DNA replication, we find no evidence of reduced progenitor proliferation, and instead we find that *Mcm5* is required for the expression of the type I BMP receptor Tkv, which is critical for the specification of the Tv4/FMRFa neuron. These results suggest that the MCM2-7 complex may play roles during CNS development outside of its well-established role during DNA replication.

## Introduction

Understanding nervous system development is critical for combating the many neurological and neuropsychiatric diseases caused by failures in this process [1]. The complexity shown by the nervous system is extreme, where thousands of neural cell subtypes need to be generated and specified through complex multi-step regulatory processes. Despite great progress in the decoding these events, the full repertoire of regulatory mechanisms underpinning nervous system development remains to be resolved.

The Apterous cluster (Ap cluster) in the *Drosophila melanogaster* central nervous system (CNS) has been an excellent model system for probing the molecular genetic mechanisms underlying neuronal subtype specification. The Ap cluster consist of four interneurons located in each thoracic hemisegment, defined by expression of the LIM-homeodomain transcription factor Apterous (Ap) and the transcription cofactor Eyes absent [2,3]. The four Ap neurons are the last-born cells of the NB5-6T neuroblast, with the last born-cell, the Tv4 neuron, expressing the FMRFamide (FMRFa) neuropeptide [4]. Forward and reverse genetic studies have uncovered a complex multi-step regulatory cascade specifying the four Ap cluster neurons [3–12]. Additionally, the final identity of these neurons is executed by terminal specification factor codes, which integrate the intrinsic transcriptional factor regulation cascades with extrinsic cues, involving retrograde BMP signaling [13].

To further probe the specification of the Ap cluster we performed an exploratory screen looking for genes displaying defects in the pattern expression of the FMRFa neuropeptide. We identified the DNA helicase gene *Minichromosome maintenance 5* (*Mcm5*), encoding a component of the hetero-hexameric MCM2-7 complex. We also find a similar phenotype in mutants of other components of the MCM2-7 complex, suggesting that the entire MCM2-7 complex is engaged in Ap cluster development. This complex is highly conserved in eukaryotes and plays a major role in DNA replication and cell cycle progression [14–17]. However, our results did not show a clear connection between *Mcm5* and progenitor proliferation, but rather reveal that it plays a more subtle role regulating the gene expression of components of the BMP signaling pathway. Hence, we describe a novel role for the gene *Mcm5*, which unexpectedly display a highly specific, and critical, role in the specification of the FMRFa neuropeptidergic neuron.

## Results

### *Mcm5* is necessary but not sufficient for Tv4 peptidergic identity specification

FMRFa has a well-studied expression pattern in the embryonic *Drosophila* ventral nerve cord (VNC), restricted to two groups of neurons: the six Tv4 neurons of the Apterous (Ap) cluster present in the thoracic hemisegments T1 to T3, generated by thoracic neuroblast 5–6 (NB5-6T), and the anterior FMRFa SE2 neurons, which are generated by a different neuroblast in

the 2nd sub-esophageal segment (Fig 1A–1C). To further probe the specification of these neurons, we performed a candidate gene screen of 35 genes, known to be expressed in the CNS at stages 11–16 (during CNS embryonic development), scoring for genes displaying changes in the FMRFa expression (Table 1). One of these genes, *Minichromosome maintenance 5* (*Mcm5*), selectively affected the proFMRFa expression in Tv4 neurons, while SE2 neuron expression was unaffected (Fig 1D, 1E, 1G and 1H). To determine if *Mcm5* mutants only affect the Tv4 neurons, or if it affects other neuropeptides express by Ap cluster, we analyzed the expression of Nplp1 which is expressed in the Tv1 neuron and in a subset of neurons medial-dorsally along the whole VNC. We did not observe any loss of Nplp1 in *Mcm5* mutants (Fig 1I and 1J).

During early stages of *Drosophila* development, *Mcm5* shows both maternal and ubiquitous zygotic expression [18,19] However, at stage 11–12 expression becomes restricted to the CNS and gut, and from stage 13–16 expression is largely confined to the CNS, where it is expressed in subsets of cells [18,19]. This selective expression raised the possibility that *Mcm5* gain-of-function could be sufficient to trigger ectopic FMRFa expression. To test this idea, we used the Gal4-UAS system to misexpress *Mcm5*, by crossing *UAS-Mcm5* to the *prospero* (*pros*) Gal4 driver, which expresses Gal4 in most, if not all neural lineages of the VNC from early stages of CNS development. However, in these misexpression embryos, we did not observe any ectopic FMRFa expression (Fig 1F–1H), indicating that although *Mcm5* is necessary, it is not sufficient for the expression of FMRFa. To ensure that the *UAS-Mcm5* transgene works correctly and to verify that the *Mcm5* phenotype can be rescued by the introduction of this *Mcm5* transgene we reintroduced *UAS-Mcm5* into the *Mcm5* mutant background (*Mcm5, prospero>UAS-Mcm5*). We found that the *UAS-Mcm5* transgene worked correctly and observed robust rescue (S1A and S1B Fig). These findings show that *Mcm5* is necessary but not sufficient for the expression of FMRFa.

## The role of *Mcm5* may be linked to the MCM2-7 complex

Mcm5 is an important component of the MCM2-7 complex [14–16]. To address if the defects observed in *Mcm5* mutants were related to its role within the MCM2-7 complex, or whether they were due to a novel and independent function of Mcm5, we analyzed mutants for all components of complex. We found that FMRFa was absent in the *Mcm4*, *Mcm5* and *Mcm7* mutants, while there was no significant effect upon FMRFa expression in the *Mcm2*, *Mcm3* and *Mcm6* mutants (S1C–S1J Fig). The phenocopy of *Mcm5*, *Mcm4* and *Mcm7* support the implication of the MCM2-7 complex in FMRFa/Tv4 generation and/or specification.

To unravel whether the lack of FMRFa is due to DNA damage caused by dysfunction of the MCM2-7 complex, we analyzed DNA damage using the Anti-H2AvD antibody in *Mcm5* mutants. We found increased H2AvD immunostaining randomly distributed across the whole CNS in *Mcm5* mutants when compared to the wild type (S2A and S2B Fig). However, we did not find H2AvD positive signal of DNA damage in the apterous cluster cells (S2C and S2D Fig).

## *Mcm5* is not required for NB5-6T lineage progression

To probe the spatiotemporal requirement of *Mcm5* during FMRFa gene expression we analyzed the expression of Mcm5. This revealed that it Mcm5 is expressed in both the NB5-6 neuroblast and in postmitotic neurons in this lineage (S2E and S2F Fig). This broad expression within the lineage did not provide any clues as when Mcm5 is necessary for the specification of the FMRFa fate. Because the MCM2-7 complex is involved in DNA replication and its mutants show defects in cell cycle progression and cell proliferation [20–24], we therefore addressed if *Mcm5* is required for the progression of the NB5-6T neuroblast lineage.

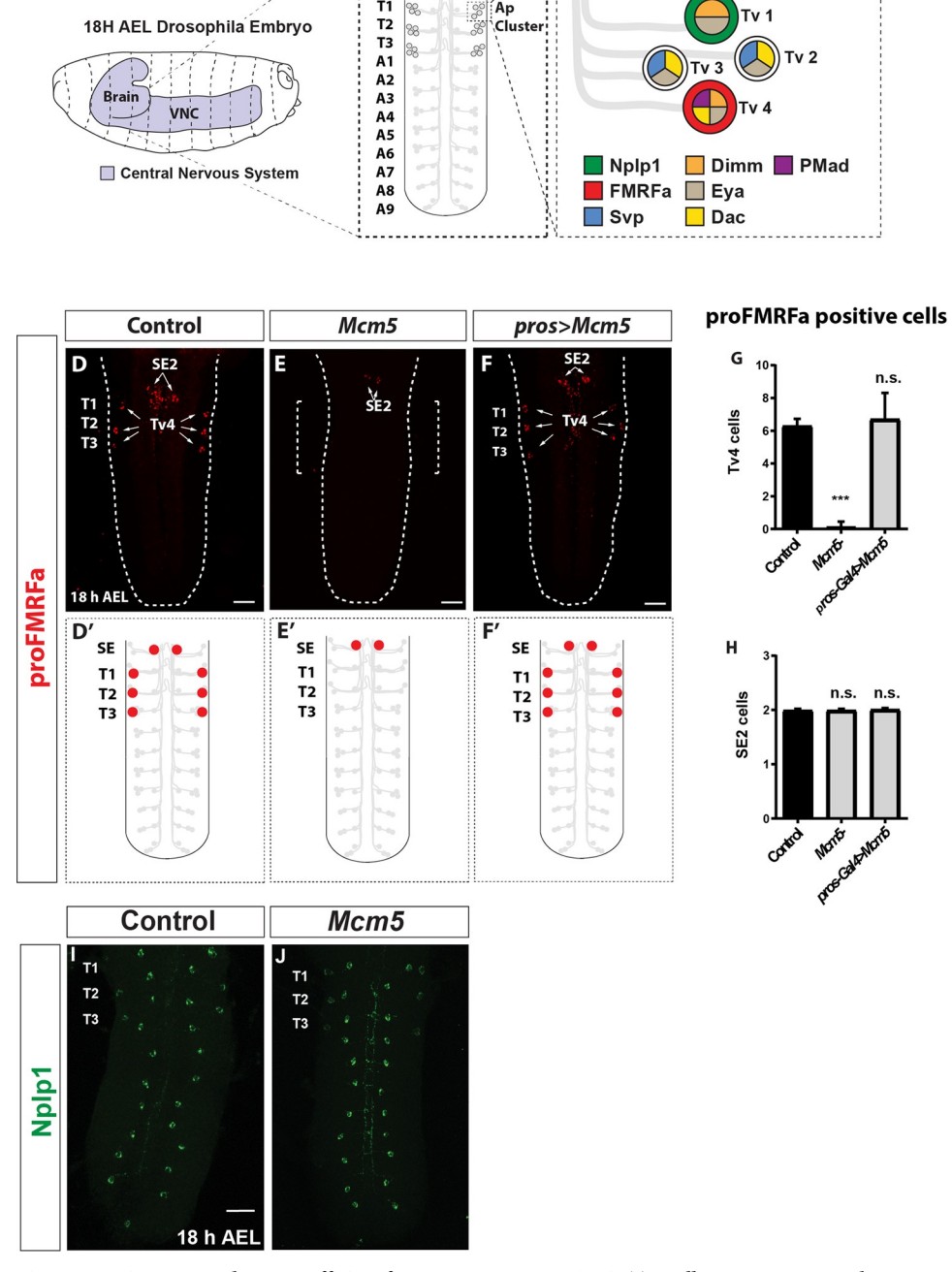

**Fig 1. *Mcm5* is necessary but not sufficient for proFMRFa expression in Tv4 cells. *Mcm7* mutants phenocopy *Mcm5* mutants.** (A-C) Schematic representation of Drosophila embryo Nervous System (A), with detailed position of the Ap cluster within the VNC (B) and some of the most representative expressed markers which identifies each Tv neuron within the Ap Cluster (C). (D-F) Immunostaining for proFMRFa in Control (D), *Mcm5* mutant (E), *Mcm5* misexpression (F), at stage 18 AEL (Scale bar 20μm). (E) Quantification of proFMRFa expressing cells within the Ap cluster in Control, *Mcm5* mutant and *Mcm5* misexpression. (G-H) Quantification of proFMRFa expressing cells within the thoracic and second suboesophageal segment (SE2) in control, *Mcm5* mutant and *Mcm5* misexpression (U-Mann-Whitney test; n≥10 CNS per genotype; *** = p<0,001). (I-J) Immunostaining for Nplp1 in Control (G) and *Mcm5* mutant (H), at stage 18h AEL. Genotypes: (A, G, I) *Oregon-R, w^{1118}*. (B, H, J) *Mcm5^{exc222}/Mcm5^{exc222}* (C) *prospero-Gal4/UAS-Mcm5*.

**Table 1. Gene screen of FMRFa altered expression pattern.** Values relative to the gene (FlyBase gene symbol) identifying the loss of FMRFa in Tv4 cells scored by immunostainig of proFMRFa in embryos at stages 11–16. It is indicated the specific allele tested as well as the fly stock origin.

| Gene | Allele | Loss of FMRFa in Tv4 cells | Stock source |
|---|---|---|---|
| alpha-Man-IIa | G4901 | None | BDSC: #30106 |
| AOX1 | lpo | None | BDSC: #37751 |
| Aplip1 | EK4 | None | BDSC: #24632 |
| babos | KG05061 | None | BDSC: #13626 |
| CG15771 | EP1405 | None | BDSC: #17004 |
| CG3967 | KG05974 | None | BDSC: #14135 |
| CG4707 | MB11625 | None | BDSC: #29232 |
| CG6023 | GG01584 | None | BDSC: #13475 |
| DIP1 | EY02625 | None | BDSC: #15577 |
| Disco | 1 | None | BDSC: #5682 |
| Drgx | GLO | None | BDSC: #56546 |
| InaE | N125 | None | BDSC: #42243 |
| Kat-60L1 | c01236 | None | BDSC: #10430 |
| Kdm4B | KO | None | BDSC: #76243 |
| Klp68D | KG03849 | None | BDSC: #13511 |
| Lmx1a | MB07364 | None | BDSC: #26346 |
| M1BP | 0932-G4 | None | BDSC: #63907 |
| Mcm5 | exc222 | Loss of cells | [38] |
| mir-278 | KO | None | BDSC: #58909 |
| Mlt | G18151 | None | BDSC: #27434 |
| Mnt | 1 | None | BDSC: #64768 |
| Mst87F | DP01376 | None | BDSC: #21844 |
| Ncd | 1 | None | BDSC: #102305 |
| Nct | J1 | None | BDSC: #63240 |
| Npc2b | 19 | None | BDSC: #41761 |
| Nrt | 2 | None | BDSC: #25033 |
| nrv2 | k04223 | None | BDSC: #102305 |
| nwk | 1 | None | BDSC: #51626 |
| Rpt6 | G1225 | None | BDSC: #33469 |
| SC35 | KG0298 | None | BDSC: #12904 |
| Sep2 | EY05856 | None | BDSC: #16680 |
| Syn | 97 | None | BDSC: #83706 |
| Syx6 | EY14508 | None | BDSC: #20941 |
| Thor | 2 | None | BDSC: #9559 |
| Tsp42Ej | 1 | None | BDSC: #27335 |

The identity of each neuron of the Ap cluster is accomplished by a complex genetic cascade [13]. To address the role of *Mcm5* within this cascade we immunostained for key Ap cluster determinants: Ap, Eya, Cas, Grh, Sqz, Nab, Dimm, Svp and Dac [3,4,10,25,26]. We did not find apparent changes in the expression of Ap, Eya, Cas, Sqz, Nab, Svp or Dac, revealing that Ap cluster cells were generated in the correct numbers (Fig 2A–2N). However, we found that *Mcm5* mutants showed Dimm expression in only one of the four Ap cluster cells: the Tv1 neuron (Fig 2G and 2H). Dimm is specifically expressed in both of the two neuropeptidergic neurons of the Ap cluster: Tv1/Nplp1 and Tv4/FMRFa. To address which cell type displayed loss of Dimm expression, we co-immunostained for Nplp1, FMRFa and Dimm in *Mcm5* mutants,

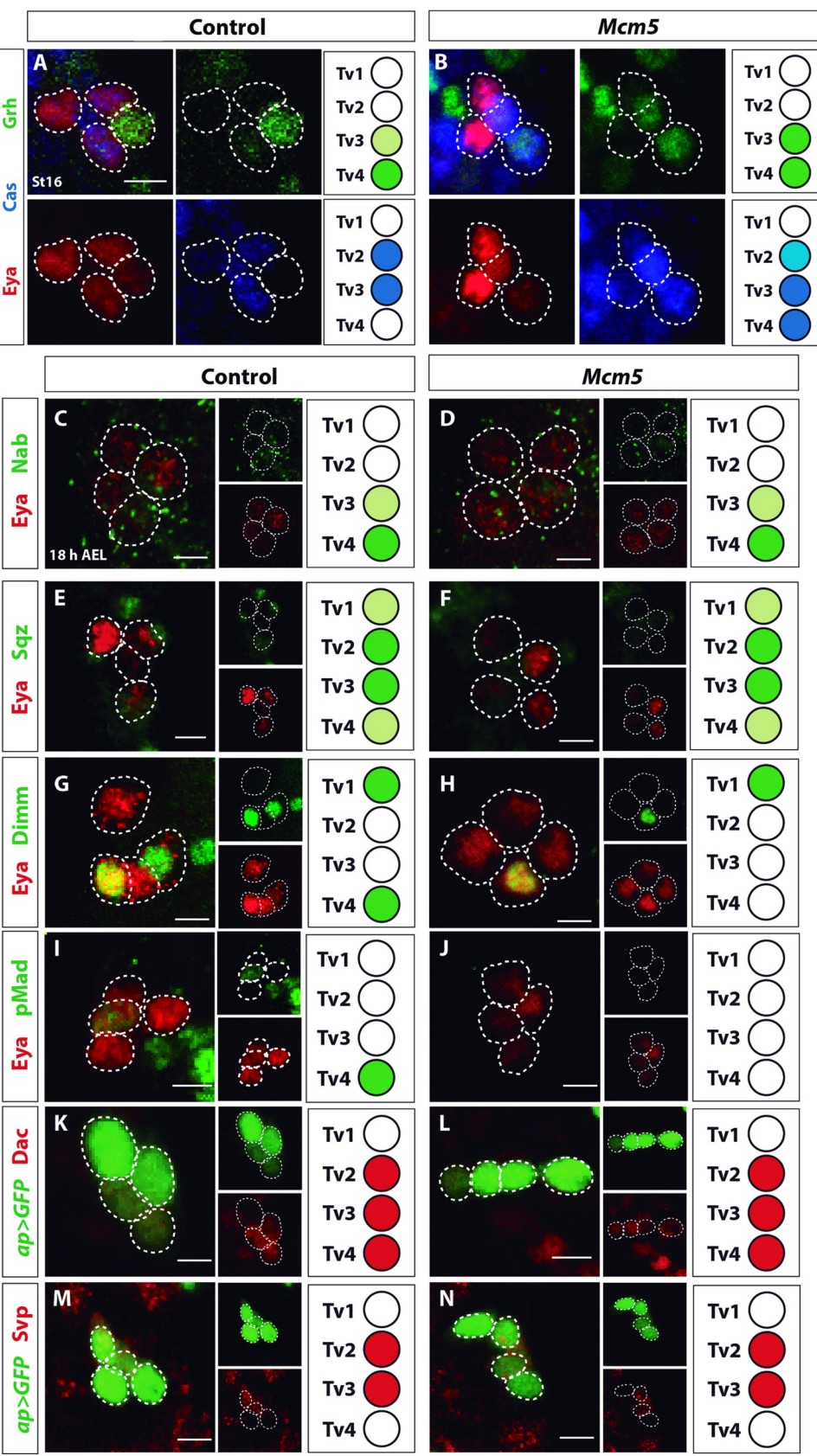

**Fig 2. Study of cell fate determinants of Apterous neuron specification in *Mcm5* mutants.** (A-B) Expression of the temporal factors Castor (Cas) and Grainy head (Grh) in the Ap cluster in control and *Mcm5* mutants, at St16. Eya was used to visualize Ap neurons. (E-J) Expression of Ap neuron determinants Nab, Squeeze (Sqz), Dimmed (Dimm) and phosphorylated form of Mothers against dpp (pMad) in control and *Mcm5* mutants, at stage 18 AEL. Eya was used to visualize Ap neurons. (K-N) Expression of *ap* based on the *ap-Gal4>UAS-GFP* reporter expression, Dachshund (Dac) and Seven up (Svp) in control and *Mcm5* mutants, at stage AFT. Merge and individual antibody images are shown in each panel. Representative images of n≥3 CNS per genotype, n≥ 18 hemisegments per genotype. Genotypes: (A, C, E, G, I, K, M) *w^1118*. (B, D, F, H, J, L, N) *Mcm5^exc222*/*Mcm5^exc222* (Scale bar 5μm).

which revealed proper specification of the Tv1 cell, evident by co-expression of Nplp1 and Dimm (S2H and S2I Fig). While Dimm is an important regulator of FMRFa expression, *dimm* mutants only display a reduction in FMRFa expression [5,27,28], not a complete loss, as is the case of *Mcm5* mutants. Hence, the loss of Dimm expression from the Tv4 cell could only partially explain the absence of FMRFa in *Mcm5* mutants.

Summarizing, since the majority of Ap cluster determinants are unaltered in *Mcm5* mutants we conclude that *Mcm5* is not required for NB5-6T progenitor proliferation, progression of the Tv4 genetic cascade or the survival of the Tv4 cell.

## BMP signaling is disrupted in *Mcm5* mutants

FMRFa expression in the Tv4 neuron is not only dependent upon an elaborate transcriptional cascade but also upon a retrograde instructive signal, provided by target-derived transforming growth factor β (TGFβ)/bone morphogenetic protein (BMP) [6,29]. Specifically, Tv4 neurons project their axons towards the midline and exit the VNC at the dorsal midline, to innervate a peripheral secretory gland; the dorsal neurohemal organ (DNH). When the axon reaches the DNH, it receives the TGFβ/BMP ligand Glass bottom boat (Gbb), which binds to a hetero-tetrameric receptor complex consisting of two receptor pairs: the type I (Saxophone; Sax and Thick veins; Tkv) and type II (Wishful thinking; Wit) BMP receptors [6,30,31]. When activated, this complex results in the phosphorylation of Mad [32,33], triggering the expression of the FMRFa neuropeptide gene [34,35]. Interestingly, while the majority of Ap cluster determinants were unperturbed, we observed loss of pMad staining in the Ap cluster neurons in *Mcm5* mutants (Fig 2I and 2J). We also analyzed the expression pattern of both pMad and Dimm in *Mcm7* mutants and observed loss of pMad and Dimm in Tv4 also in *Mcm7* mutants (S2J and S2K Fig). These results suggest that the lack of FMRFa expression in Tv4 in *Mcm5* and *Mcm7* mutants could be due to the disruption of the TGF-β/BMP signaling pathway.

## Expression of type I BMP receptors rescues FMRFa expression

The disruption of TGF-β/BMP signaling in the Tv4 neuron can have several underpinnings: 1) a failure of Tv4 axon pathfinding to the DNH, 2) a lack of DNH development, 3) a lack of BMP ligand (Gbb) expression in the DNH, and/or 4) a defect in TGF-β/BMP signaling in the Tv4 neuron itself [6]. Previous studies have demonstrated that the first three problems can be rescued by the expression of the Gbb ligand [6]. On this note, to probe if anyone of these three first mentioned phenotypes may underlie the loss of FMRFa in *Mcm5* mutants, we used *pros-Gal4* to express *UAS-gbb* in the developing CNS. However, expression of the Gbb ligand did not result in any rescue of FMRFa in *Mcm5* mutants (Fig 3A–3C and 3F).

The fourth potential underpinning of the loss of FMRFa expression in *Mcm5* mutants could be a defect in TGF-β/BMP signaling in the Tv4 neuron itself. To probe this issue, we used *pros-Gal4* to express activated versions of the type I BMP receptors *sax* and *tkv* (*UAS-saxA* and *UAS-tkvA*) [6,12,30,36]. Intriguingly, we observed robust rescue of FMRFa (Fig 3D and 3F). Additionally, in few cases we also observed ectopic expression of FMRFa in the Ap

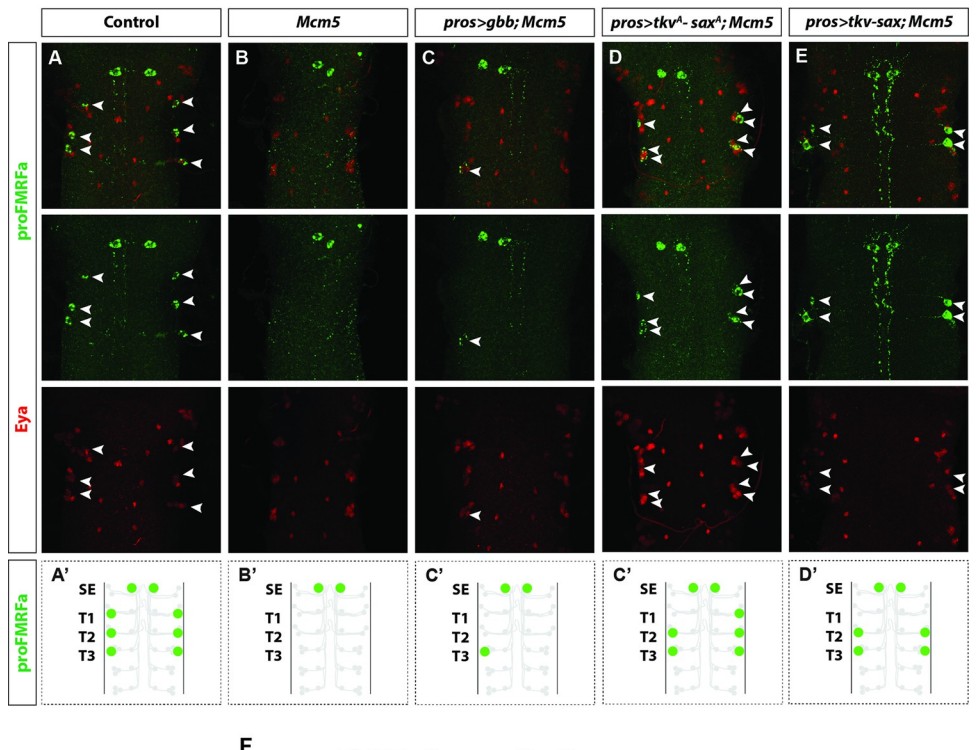

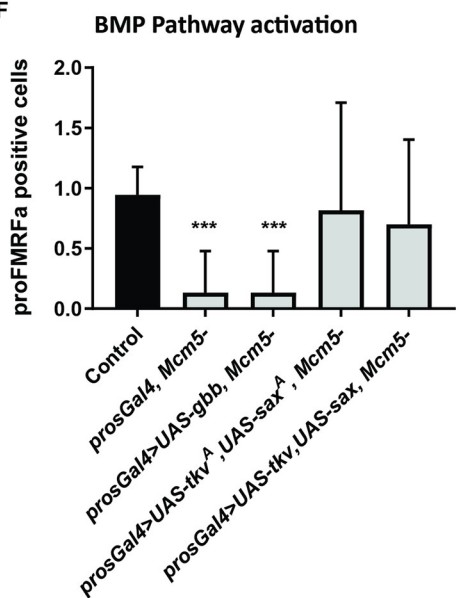

**Fig 3. BMP receptor transgenic expression rescues FMRFa expression in *Mcm5* mutants.** (A-E) Expression of Eya and proFMRFa, in control (A), *Mcm5* mutants (B), expression of Gbb ligand (*gbb*) in *Mcm5* (C), expression of the activated type I BMP receptors (*sax$^A$* and *tkv$^A$*) in *Mcm5* mutants (D), expression of wild type forms of the type I BMP receptors (*sax* and *tkv*) in *Mcm5* mutants (E), at stage 18h AEL (thoracic segments T1-T3). (F) Quantification of proFMR expressing cells (Tv4) in control, *Mcm5* mutants and rescue genotypes (U-Mann-Whitney test; n≥6 CNS per genotype, n≥ 30 hemisegments per genotype, *** = p<0.001; n.s = non-significant). Merge and individual antibody images are shown in each panel. Genotypes: (A) *Oregon-R, w$^{1118}$*. (B) *pros-Gal4; Mcm5$^{exc222}$*. (C) *pros-Gal4/UAS-gbb; Mcm5$^{exc222}$* (D) *pros-Gal4/UAS-saxA, UAS-tkvA; Mcm5$^{exc222}$*. (E) *pros-Gal4/UAS-sax, UAS-tkv; Mcm5$^{exc222}$*.

cluster (Fig 3D and 3F). To further probe for defects in TGF-β/BMP signaling, we also tried rescuing FMRFa using the wild type forms of the type I BMP receptors (*UAS-sax* and *UAS-tkv*) in *Mcm5* mutants. This also resulted in robust rescue of FMRFa (Fig 3E and 3F). Additionally, we verified that neurons which had rescued FMRFa using the wild type forms of the type I BMP receptors also restored pMad expression (S3C and S3D Fig), indicating that BMP TGF-β/BMP signaling is restored.

The loss of pMad in the Tv4 neuron, coupled with the failure of Gbb to rescue while expression of either of the type I BMP receptors, as wild type or constitutively activated versions, rescues the expression of FMRFa, indicates defects in TGF-β/BMP signaling in *Mcm5* mutants.

## RNA-seq analysis of *Mcm5* mutants shows an altered expression of the BMP receptor gene *thickveins*

Why is BMP signaling perturbed in *Mcm5* mutants? To unravel which specific genes were responsible of this interruption we analyzed *Mcm5* mutant gene expression through RNA-seq analysis of RNA extracted from control and *Mcm5* mutant embryos, at St16-18hAEL. We detected the expression of 21,075 gene isoforms, of which 244 were significantly up-regulated and 171 significantly down-regulated (Fig 4A and 4B and S1 Table). Therefore, only 2,78% of the detected genes showed significantly altered expression in *Mcm5* mutants. To identify if the genes dysregulated in *Mcm5*mutants were related or belonged to a specific biological process, we performed gene ontology enrichment analysis. Analysis of the 415 affected genes did not reveal any significative differences (Benjamini-Hochberg, p<0.005) for "biological process" or "molecular component", indicating that the genes dysregulated in *Mcm5* do not show an obvious relationship.

Focusing on TGF-β/BMP pathway components, we observed a strikingly low expression level (RPKM) for *tkv* in *Mcm5* mutants, when compared to the control, while other TGF-β/BMP components were unaffected. To shed more light upon the altered expression of *tkv* we analyzed the expression of individual *tkv* exons. We observed a striking reduction in read coverage for exons 1 and 2 of the *tkv* gene (Fig 4C, top). These exons are the only exons exclusive for *tkv* transcription products, with other exons sharing overlapping sequences related to other genes and non-coding RNAs transcripts (Fig 4C bottom, Flybase).

To elucidate if the downregulation of *tkv* expression in *Mcm5* mutants generally affects TGF-β/BMP signaling in the VNC or is confined to specific cell types, we analyzed pMad immunostaining expression pattern in the entire VNC. This did however not reveal any obvious differences in the overall distribution of pMad (S3A and S3B Fig). This indicated that the downregulation of *tkv* expression only affects a minor sub-set of pMad positive cells in the VNC.

Summarizing, the RNA-seq data supports the results obtained from marker expression and genetic analysis, strongly suggesting that the reduction of *tkv* expression in *Mcm5* mutants is responsible for the defect in TGF-β/BMP signaling, and thereby the loss of FMRFa expression.

## Discussion

### Novel function of the *Mcm5* gene

This study reveals a novel function for *Mcm5*, acting as a specific factor for Tv4 neuropeptidergic identity. Initially, we envisioned that the observed phenotype, loss of FMRFa, could be explained by the already known function of the MCM2-7 complex in DNA replication and cell proliferation [15,23,24]. This could have manifested as a blocked NB5-6T lineage progression and a failure in generating the last-born neuron in the lineage; the Tv4/FMRFa neuron.

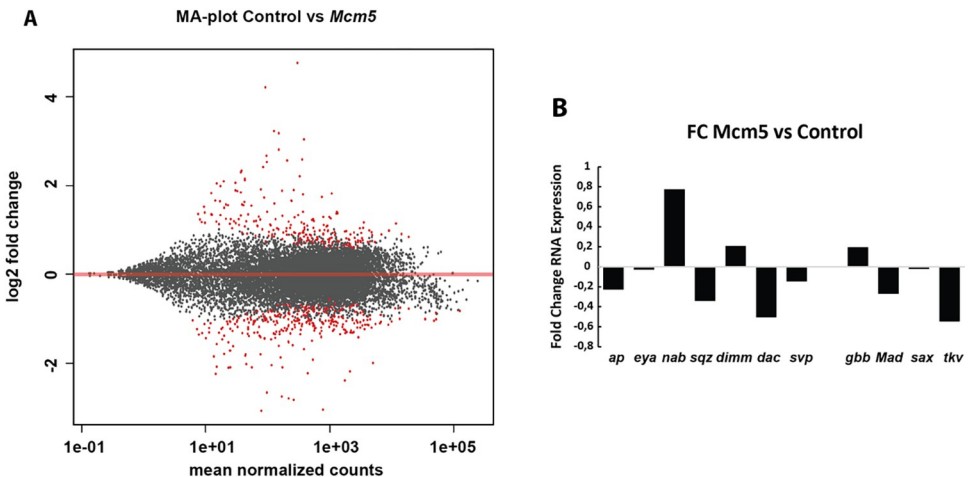

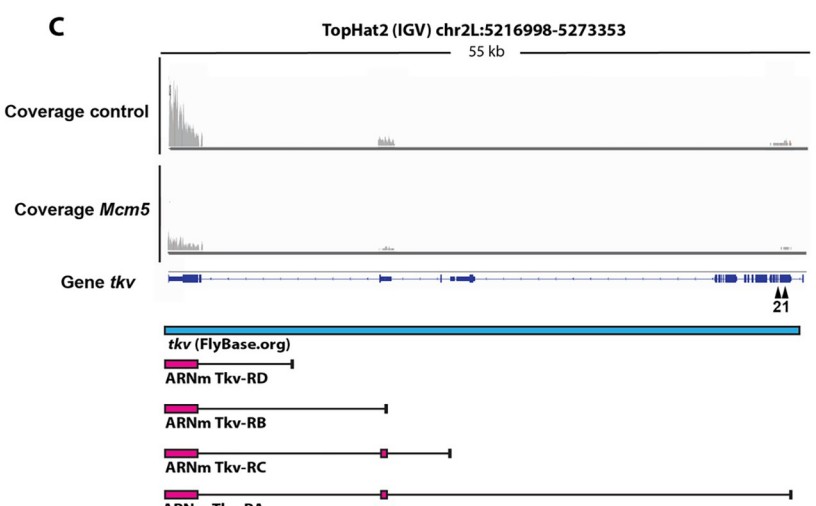

**Fig 4. RNA-seq analysis reveals downregulation of TGFb/BMP pathway genes in *Mcm5* mutants.** (A) Bland-Altman graph showing differences in the expression of every gene analyzed through RNA-seq from control and *Mcm5* mutants in St16-AFT embryos. The expression of every gene is represented by a dot. Dots are located in the graph according to change in expression between conditions in a logarithmic scale, when compared with the media of the normalized expression. Genes with differences in expression between genotypes are represented in red, otherwise they are grey. Genes represented in positive values show higher expression in *Mcm5* mutants than in controls and *vice versa* (graph made with DESeq2, statistical analysis performed by hypergeometric distribution adjusted by Benjamini-Hochberg; p-value < 0,05). (B) Fold change histogram of the relative expression in *Mcm5* mutants compared to the control of representative genes involved in Tv4 specification, *ap*, *eya*, *nab*, *sqz*, *dimm*, *dac*, *svp*, and genes involved in BMP signaling pathway, *gbb*, *Mad*, *sax* and *tkv*. (C) Alignment distribution of reads obtained after RNA-seq analysis of *tkv* gene in control and *Mcm5* mutants. The top of the graph shows the enrichment in sequences corresponding to every exonic region and the gene sequence of *tkv* gene represented in indigo (TopHat2) and blue (Flybase). A striking reduction in read coverage for the exons 1 and 2 (black arrows) of the *tkv* gene in *Mcm5* mutants comparing to the control is shown. On the bottom, the graph shows the different isoforms of *tkv* RNAm represented in magenta, other genes located in the same chromosome region in green (its RNAm in yellow), and non-coding RNAs in orange (Flybase) (graph made with TopHat2, IGV). Genotypes: (A, B) $w^{1118}$ and $Mcm5^{exc222}/Mcm5^{exc222}$.

However, our marker analysis show that this is not the case and that the Ap cluster is correctly generated. Indeed, the other peptidergic neuron of the Ap cluster (Tv1) is specified and expresses its corresponding neuropeptide (Nplp1). Detailed analysis of several known factors in the FMRFa specification shows that the expression of Dimm is absent in the Tv4 neuron.

However, because *dimm* mutants only display a reduction of FMRFa expression [6,27,28], the lack of Dimm expression cannot explain the phenotype of the complete FMRFa absence observed in *Mcm5* mutants. Instead, the loss of pMad in the Tv4 neuron pointed to defects in TGF-β/BMP retrograde signaling in the Tv4 neuron. While several defects could potentially underlie the loss of pMad i.e., axon pathfinding defects, loss of Gbb expression and/or target organ development defects, our genetic rescue experiments and RNA-seq analysis both point to that the down-regulation of *tkv* underlies the loss of FMRFa expression observed in *Mcm*5 mutants. Furthermore, the specificity of the phenotype in the Tv4 neuron is reinforced by the lack of effect on the SE2 neurons, in which the neuropeptide FMRFa is properly expressed. Finally, the phenocopy observed in the *Mcm*7 and *Mcm4* mutants strongly suggests that the lack of FMRFa expression observed in the *Mcm5* mutants depends on its role within the MCM2-7 complex.

## Unraveling the "MCM paradox"

DNA replication in eukaryotes is a highly controlled process that involves various components. A central player in this process is the MCM (Minichromosome Maintenance) complex. The MCM complex is composed of a hetero-hexamer complex of six related subunits, MCM2-7, which are all highly conserved throughout evolution [21]. Despite their importance in replication, development and disease, the precise mechanism of action of MCM2-7 proteins is poorly understood. The expression levels of these proteins are much higher than what one might expect for its known functions, which is often referred to as overabundance. Furthermore, their expression patterns are not restricted to the origins of replication. The apparently excess production of the MCM complex and its expression outside of the origin of replication has resulted in what is known as the "MCM paradox" [37], and may suggest roles for the MCM complex outside DNA replication.

In the current study we reveal that *Mcm5* is involved in a highly specific neuronal differentiation process, which does not appear generally connected to the DNA replication process itself. Rather, during CNS development MCM2-7 may act in a highly specific and coordinated manner, possibly by unwinding target DNA so that transcription factors responsible for subtype specification phenomena can access DNA. In addition, our data on *Mcm7* and *Mcm4* suggest that this selective role may engage the entire MCM2-7 complex. This work shows that, rather than an independent function for *Mcm5*, its role in specification is most likely related to its helicase activity within the MCM2-7 complex, since we observed a phenocopy of the absence of FMRFa in *Mcm7*, *Mcm4* mutants (in which the helicase function of MCM2-7 is eliminated [20,21]. We suggest that the MCM complex may be involved in many other similar specific processes, which would shed further light upon the MCM paradox.

## Role of MCM during animal development

The MCM2-7 genes are linked to and drive several cancer types in humans, and when mutated in mice result in growth retardation and mortality. All six MCM2-7 genes are dynamically expressed during development, often connected to regions of high proliferation, such as the embryonic forebrain (telencephalon). However, their role in the CNS is not clear. For *Mcm5*, mice mutant for intron 15, likely hypomorphic, are associated with mortality and aging. However, there has been no *Mcm5* null analysis in mouse. *Mcm5* is dynamically expressed in *Drosophila* as well, and *Drosophila Mcm5* mutants die at L3 with absent imaginal discs and an undergrown CNS [38]. The latter phenotype contrasts with our findings in the embryo, where we did not observe any reduction in growth of the CNS, pointing to different roles for *Mcm5* in the embryonic versus larval CNS.

Recently, *MCM5* was also identified as likely causative in the human Meier-Gorlin Syndrome, a rare developmental disorder characterized by growth retardation [39]. Several other DNA replication complex genes have also been identified as causative in Meier-Gorlin syndrome, including *ORC1*, *ORC4*, *ORC6*, *CDT1*, *CDC6* and *CDC45*, members of the pre-replication (pre-RC) and pre-initiation (pre-IC) complexes, as well as *GMNN*, a regulator of cell-cycle progression and DNA replication [40]. Similar to the undergrown CNS observed in *Drosophila* Mcm5 mutants [38], 43% of human Meier-Gorlin patients display microcephaly [41]. However, the *MCM5* mutant patient identified thus far (p.T466I) did not display apparent cognitive deficits or microcephaly [39]. The *Mcm5* and *Mcm7* genes are highly conserved, and the proteins show 83% and 82% similarity between Drosophila and humans, respectively. These studies, combined with the results presented here for Mcm5 and Mcm7 in the *Drosophila* CNS, makes the case for further studies of these complexes during CNS development.

## Methods

### Fly stocks

Fly stocks were raised, and crosses were performed at 25˚C on standard medium. The following fly mutant alleles were used: $Mcm5^{exc222}$ is a null allele of *Mcm5* created by imprecise excision [38]. Homozygous mutants die in the third larval stage with a normal size. However, in this third larval stage they show small brains and lack imaginal discs. The polytene chromosomes of the salivary glands are completely normal [38] e, $Mcm2^{MI06343}$ (BL-43056), $Mcm3^{smu}$ [42], $dpa^1$ (BL-4126, *dpa* = *Mcm4*, $Mcm6^{K1214}$ (BL-4322) $Mcm7^{f03462}$ (BL-18656), $ap^{md544}$ (referred to as *apGal4*, BL-3041), *prospero-Gal4* [43], *UAS-nls-myc-EGFP*, *UAS-gbb*, $UAS-sax^a$, $UAS-tkv^a$ [6] *and UAS-sax*, *UAS-tkv* [30], *UAS-Mcm5* (FlyORF-F001278). Mutants were kept over *CyO*, *Dfd-EYFP* or *TM6*, *Sb*, *Tb*, *Dfd-EYFP* balancer chromosomes. As wild type, *Oregon-R $w^{1118}$* was often used. Unless otherwise stated, flies were obtained from the Bloomington Drosophila Stock Center.

### Immunohistochemistry

The antibodies used were: guinea pig anti-Col (1:1000); guinea pig anti-Dimm (1:1000); rabbit anti-proFMRFa (1:1000) [7]; mAb anti-Dac (1:25) (from Developmental Studies Hybridoma Bank, Iowa City, IA, US); mouse anti-Seven up (1:50) (Y. Hiromi, National Institute of Genetics, Mishima, Japan); rabbit anti-pMad (1:500) (41D10, Cell Signaling Technology); mouse mAb Anti-Eya10H6 (1:200) (from Developmental Studies Hybridoma Bank, Iowa City, IA, US); guinea pig anti-Cas (1:1000) [44]; rat anti-Sqz (1:750) [45]; rat anti-Grh (1:1000) [4]; rabbit anti-Nab (1:1000) [25], rabbit Anti-Histone H2AvD phosphoS137 (1,500) (from Rockland). Mcm5 antibody were raised in Guinea Pigs against the nh2- C+SSNKSAPSEPAEGEI–conh2w and nh2- C+FGRWDDTKGEENIDF–conh2 peptides (Cultek, Madrid, Spain). All polyclonal sera were pre-absorbed against pools of early embryos. Secondary antibodies were conjugated with Alexa Fluor 488, Rhodamine-RedX or Alexa Fluor 647, and used at 1:500 (Jackson ImmunoResearch, PA, US). Embryos were dissected in PBS, fixed for 20 minutes in 4% PFA, blocked and processed with antibodies in PBS with 0.2% Triton-X100 and 3% bovine serum albumin. Slides were mounted with Vectashield (Vector, Burlingame, CA, US). In all cases wild-type and mutant embryos were stained and analyzed on the same slide.

### Confocal imaging, data acquisition and staining quantification

Zeiss META 510 and 710 Confocal microscopes were used to collect data for all fluorescent images; confocal stacks were merged using LSM software or Adobe Photoshop CS4. Where

appropriate, images were false colored to assist color-blind readers, or to more clearly represent the data.

## Statistical methods

Statistical analysis was performed using Microsoft Excel and IBM SPSS v.26. Quantifications of observed phenotypes were performed using U-Mann-Whitney test.

## RNA-sequencing and differential expression analysis

Three biological replicates of frozen collections of St16-AFT embryos control ($w^{1118}$), $Mcm5^{exc222}$ (50 mg) were used for RNA extraction, using RNeasy Mini Kit (Qiagen, Hilden, Germany). Sample RNA yield was measured with a NanoDrop, precipitated in ethanol, and then sent to The Genomics Unit of the Madrid Science Park (Madrid, Spain) for library preparation and sequencing. Yield was checked, upon receipt of each sample, by use of NanoDrop, Qubit RNA Assay and Agilent Bioanalyzer. The samples were fragmented after RNA QC, reverse transcribed with random primers, and barcode tagged. Sequencing was performed by on the Illumina Hiseq 2500, in 1x75 bp single-read sequencing configuration (the output was stored as FASTQ-files,), which yielded 25–30 million reads/sample. The FASTQ-files were aligned against the dmel reference genome (BDGP R5/dm3 Apr. 2006) and raw reads were normalized as reads per kilobase-length of gene per million mapped sequence reads (RPKM). Raw RNA-Seq reads were aligned to *Drosophila melanogaster* (dm3) genome using TopHat (version 2.1.1) [46] with default settings, and only uniquely mapped reads were retained to compute the number of reads for exons and exon-exon junctions in each sample using the Python package HTSeq, with the annotation of the UCSC Ensembl gene annotation (dm3_ensGene) [47]. The package DESeq2 (Galauxy Version 2.11.40.2) [48] was used to formulate the counts of the reads that were aligned to each isoform of each event and the differential expression of transcripts between control and $Mcm5^{exc222}$ mutants. The Benjamini-Hochberg procedure was applied to calculate the adjusted *p*-values in the likelihood ratio test [49]. Gene Ontology enrichment analysis was performed using GeneCodis3 for GO Biological process, GO Molecular function, and GO cellular component; lowest annotation levels, Hypergeometric statistical test, FDR P-value correction [50–52].

## Supporting information

**S1 Fig. Rescue of *Mcm5* mutant phenotype by the expression of Mcm5. FMRFa expression analysis in Mcm2-7 Complex gene related mutants.** Immunostaining for proFMRFa and Eya in Control (A) and *Mcm5*, *prospero>UAS-Mcm5* (B) at stage 18h AEL. Immunostaining for proFMRFa in Control, and mutants for the Mcm2-7 complex components Mcm2 (D), Mcm3 (E), Mcm4 (F), Mcm5 (G), Mcm6 (H), and Mcm7 (I) and the respective quantification of Tv4 FMRFa positive cells (J) Merge and individual antibody images are shown in each panel. Genotypes: (A, C) $w^{1118}$. (B) *prospero-Gal4/ UAS-Mcm5; Mcm5^{exc222}*, (D) $Mcm2^{MI0634}$ (E) $Mcm3^{smu}$ (F) $dpa^1$ (G) $Mcm5^{exc222}$ (H) $Mcm7^{f03462}$.
(TIF)

**S2 Fig. DNA damage in *Mcm5* mutants. Mcm5 expression in NB5-6 Lineage. Nplp1 expression and quantification of Eya expressing cells in control and *Mcm5* mutants. Dimm and pMad expression in *Mcm7* mutants.** Immunostaining for cH2aX, Eya and proFMRFa in Control (A) and *Mcm5* mutants (B) at stage 18h AEL (scale bar 50 μm). Detail of immunostaining for cH2aX, Eya and proFMRFa in the apterous cluster cells of Control (C) and *Mcm5* (D) mutants (scale bar 5 μm). (E) Expression of Mcm5 protein detected by immunostaining in

the NB 5–6, identified by the reporter line *Lbe(k)-GFP*, and Deadpan immunostaining (Dpn). (F) Expression of Mcm5 protein detected by immunostaining in the Ap cluster cells, identified by Eya. (G) Quantification of Eya expressing cells within thoracic segments T2 and T3 in control and *Mcm5* mutants (U-Mann-Whitney test; n≥3 CNS per genotype, n≥ 9 hemisegments per genotype, n.s = non-significant). (H-I) Immunostaining for proFMRFa, Nplp1 and Dimm in Control (H) and *Mcm5* mutant (I), at stage 18h AEL (scale bar 5 μm). (J-K) Immunostaining for proFMRFa, Eya and pMad in Control (J) and *Mcm7* mutant (K), at stage 18h AEL (scale bar 5 μm). Merge and individual antibody images are shown in each panel. Genotypes: (A, C, E, F, H, J) *w^1118^*. (B, D, I) *Mcm5^exc222^/Mcm5exc222* (K) *Mcm7^f03462^/Mcm7^f03462^*. *Mcm3^smu^* (F) *dpa^1^* (G) *Mcm5^exc222^* (H) *Mcm7^f03462^*.
(TIF)

**S3 Fig. Immunostaining for p-Mad in *Mcm5* mutants.** Immunostaining for p-Mad in Control (A) and *Mcm5* mutants (B) at stage 18h AEL. (C) Quantification of Cas positive Ap cluster cells in control and *mcm5* mutant (U-Mann-Whitney test; n≥3 CNS per genotype, n≥ 18 hemisegments per genotype, n.s = non-significant). Immunostaining for proFMRFa, Eya and pMad in Control (D) and expression of wild type forms of the type I BMP receptors (*sax* and *tkv*) in *Mcm5* mutants (E), at stage 18h AEL. Genotypes: (A,D) *w^1118^* (B) *Mcm5^exc222^/Mcm5^exc222^*, (E) *pros-Gal4/UAS-sax*, *UAS-tkv*; *Mcm5^exc222^*.
(TIF)

**S1 Table. Gene expression levels in *Mcm5* mutants.** Values relative to the gene (FlyBase gene symbol) expression levels in three replicates of each control (W1, W2, W3) and *Mcm5* mutant (MCM51, MCM52, MCM53) expressed as FRPKM (fragments per kilobase per million mapped reads). DESeq2 analysis values for fold change (FC), p-value, and Benjamini-Hochberg adjusted p-value (p-adj) are indicated for each gene.
(PDF)

**S2 Table. Numerical datasets that underlie graphs and statistics.**
(XLSX)

## Acknowledgments

We are grateful to Y. Hiromi, F. Díaz-Benjumea, R.S. Hawley and the Developmental Studies Hybridoma Bank at the University of Iowa, and The Bloomington Stock Center for sharing antibodies, fly lines and DNAs.

## Author Contributions

**Conceptualization:** Irene Rubio-Ferrera, Stefan Thor, Jonathan Benito-Sipos, Ignacio Monedero Cobeta.

**Data curation:** Irene Rubio-Ferrera, Jonathan Benito-Sipos.

**Formal analysis:** Irene Rubio-Ferrera, Pablo Baladrón-de-Juan, Luis Clarembaux-Badell, Ignacio Monedero Cobeta.

**Funding acquisition:** Jonathan Benito-Sipos.

**Investigation:** Irene Rubio-Ferrera, Pablo Baladrón-de-Juan, Luis Clarembaux-Badell, Marta Truchado-Garcia, Sheila Jordán-Álvarez.

**Methodology:** Jonathan Benito-Sipos.

**Project administration:** Jonathan Benito-Sipos, Ignacio Monedero Cobeta.

**Supervision:** Jonathan Benito-Sipos, Ignacio Monedero Cobeta.

**Visualization:** Irene Rubio-Ferrera, Pablo Baladrón-de-Juan, Luis Clarembaux-Badell, Stefan Thor, Ignacio Monedero Cobeta.

**Writing – original draft:** Jonathan Benito-Sipos, Ignacio Monedero Cobeta.

**Writing – review & editing:** Irene Rubio-Ferrera, Stefan Thor, Jonathan Benito-Sipos, Ignacio Monedero Cobeta.

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
