## [Decision Letter · Decision Letter 0]

6 Dec 2021

Dear Dr Monedero Cobeta,

Thank you very much for submitting your Research Article entitled 'Selective role of the DNA helicase Mcm5 in BMP retrograde signaling during Drosophila neuronal differentiation' to PLOS Genetics.

The manuscript was fully evaluated at the editorial level and by independent peer reviewers. The reviewers appreciated the attention to an important problem, but raised some substantial concerns about the current manuscript. Based on the reviews, we will not be able to accept this version of the manuscript, but we would be willing to review a much-revised version. We cannot, of course, promise publication at that time.

If you decide to revise the manuscript for further consideration at PLOS Genetics, please aim to resubmit within the next 60 days, unless it will take extra time to address the concerns of the reviewers, in which case we would appreciate an expected resubmission date by email to plosgenetics@plos.org.

[LINK]

We are sorry that we cannot be more positive about your manuscript at this stage. Please do not hesitate to contact us if you have any concerns or questions.

Yours sincerely,

Hongyan Wang, Ph.D.

Associate Editor

PLOS Genetics

Gregory P. Copenhaver

Editor-in-Chief

PLOS Genetics

Reviewer's Responses to Questions

**Comments to the Authors:**

Reviewer #1: This study by Irene Rubio-Ferrera has revealed an unexpected and rather specific role for the MCM2-7 complex in neural development. Instead of affecting stem cell cycling as expected from its known role in unwinding DNA at the replicative fork during DNA replication, Irene found in both Mcm5 and Mcm7 mutants a specific neuronal cell fate change in a well characterized neuronal lineage. Irene further ascribed this limited fate defect to loss of expression of the type I BMP receptor Tkv. This study is beautifully done. It lends further support for the much less appreciated role of MCM complex in gene expression.

Specific concern: It is unclear exactly where and when Mcm5 acts to ensure the correct Tv4/FMRFa neuron fate. It was mentioned about Mcm5 expression in subsets of CNS cells, which is interesting and should be mapped with respect to the neuronal lineage of interest. Further, one should attempt rescue with Mcm5 transgene (in addition to the Mcm5 misexpression) and (if possible) more targeted Mcm5 knockdown.

General point: It was only mentioned near the end of Discussion about the undergrown CNS observed in Drosophila Mcm5 mutants. This info is highly relevant and should be made aware of upon mentioning of Mcm5 mutants. One should also try to reconcile the limited phenotype observed in this study with the undergrown CNS phenotype reported before.

Reviewer #2: In this manuscript, Rubio-Ferrera et al. screened genes required for CNS development by using the Tv4 neuron of the fly ventral nerve cord as a model. They identified Mcm5, loss of which led to a neurospecification defect of Tv4 neuron. They went on to test if Mcm5 is required for the progression of NB5-6T lineage, and found no obvious change in Ap cluster determinants. Further, they found that Mcm5 is required for BMP signaling in Tv4 neurons. Transgenic expression of pros>saxA or tkvA rescues Mcm5 mutant, suggesting Mcm5 functions through BMP signaling. Lastly, they performed RNA-seq of Mcm5 mutant embryos, and found that tkv expression is low in Mcm5 mutant, supporting their markers expression study. Overall, this is an interesting study. The genetic experiments clearly suggest that Mcm5 plays a role in brain development, albeit the detailed mechanism how Mcm5 regulates BMP signaling awaits investigation. Also, some control experiments are missing.

Is the novel role the Mcm5 helicase plays in neuronal differentiation specific to Mcm5 (and Mcm7)? To claim that the whole complex Mcm2-7 is involved in neuronal subtype specification, I wonder if knockdown of the other 4 Mcm genes gives rise to the same phenotype in the specification of Tv4 neuron?

Mcm7 phenocopies Mcm5. Is Mcm5 and Mcm7 functionally interchangeable in regulating neuronal differentiation?

Is Mcm7 also required for the expression of Tkv or other BMP signaling components? Could pros>saxA or tkvA rescues Mcm7 defect?

As controls, can transgenic pros>Mcm5 expression rescue the Mcm5 defect in Fig1 and Fig3?

Also a bit of a stretch, to relate the study with mammalian CNS development, could human Mcm5 or 7 transgenes rescue the defects in fly brain?

Also, it would be nice of the list and the results of the 35 candidates are revealed (maybe as a supplement table).

Reviewer #3: My review is uploaded as an attachment.

**Have all data underlying the figures and results presented in the manuscript been provided?**

Reviewer #1: None

Reviewer #2: Yes

Reviewer #3: Yes

PLOS authors have the option to publish the peer review history of their article (what does this mean?). If published, this will include your full peer review and any attached files.

Reviewer #1: No

Reviewer #2: No

Reviewer #3: No

---

## [Decision Letter · Decision Letter 1]

29 Mar 2022

Dear Dr Monedero Cobeta,

Thank you very much for submitting your Research Article entitled 'Selective role of the DNA helicase Mcm5 in BMP retrograde signaling during Drosophila neuronal differentiation' to PLOS Genetics.

The revised manuscript was fully evaluated at the editorial level and by independent peer reviewers. Although two reviewers are satisfactory with the revision, reviewer 3 stilled raised substantial concerns on the manuscript. The Editors share the same concern with reviewer 3 that the molecular mechanisms presented in this manuscript is weak. We will not be able to accept this version of the manuscript, but we would be willing to review a much-revised version in which additional data are provided to support the conclusion. We cannot, of course, promise publication at that time.

If you decide to revise the manuscript for further consideration at PLOS Genetics, please aim to resubmit within the next 60 days, unless it will take extra time to address the concerns of the reviewers, in which case we would appreciate an expected resubmission date by email to plosgenetics@plos.org.

[LINK]

We are sorry that we cannot be more positive about your manuscript at this stage. Please do not hesitate to contact us if you have any concerns or questions.

Yours sincerely,

Hongyan Wang, Ph.D.

Associate Editor

PLOS Genetics

Gregory P. Copenhaver

Editor-in-Chief

PLOS Genetics

Reviewer's Responses to Questions

**Comments to the Authors:**

Reviewer #1: Well done

Reviewer #2: It would be nice if the protein similarity of Mcm5 and Mcm7 genes between Drosophila and humans are explicitly described.

Reviewer #3: My review has been uploaded as an attachment.

**Have all data underlying the figures and results presented in the manuscript been provided?**

Reviewer #1: None

Reviewer #2: Yes

Reviewer #3: Yes

PLOS authors have the option to publish the peer review history of their article (what does this mean?). If published, this will include your full peer review and any attached files.

Reviewer #1: No

Reviewer #2: No

Reviewer #3: No

---

## [Decision Letter · Decision Letter 2]

13 May 2022

Dear Dr Monedero Cobeta,

We are pleased to inform you that your manuscript entitled "Selective role of the DNA helicase Mcm5 in BMP retrograde signaling during Drosophila neuronal differentiation" has been editorially accepted for publication in PLOS Genetics. Congratulations!

Yours sincerely,

Hongyan Wang, Ph.D.

Associate Editor

PLOS Genetics

Gregory P. Copenhaver

Editor-in-Chief

PLOS Genetics

PLOS authors have the option to publish the peer review history of their article (what does this mean?). If published, this will include your full peer review and any attached files.

**Data Deposition**

http://datadryad.org/submit?journalID=pgenetics&manu=PGENETICS-D-21-01485R2

**Press Queries**

---

## [Editor Report · Acceptance letter]

17 Jun 2022

PGENETICS-D-21-01485R2 

Selective role of the DNA helicase Mcm5 in BMP retrograde signaling during Drosophila neuronal differentiation 

Dear Dr Monedero Cobeta, 

We are pleased to inform you that your manuscript entitled "Selective role of the DNA helicase Mcm5 in BMP retrograde signaling during Drosophila neuronal differentiation" has been formally accepted for publication in PLOS Genetics! Your manuscript is now with our production department and you will be notified of the publication date in due course.

With kind regards,

Anita Estes

PLOS Genetics

On behalf of:
